# Approaching Polycarbonate as an LFT-D Material: Processing and Mechanical Properties

**DOI:** 10.3390/polym15092041

**Published:** 2023-04-25

**Authors:** Christoph Schelleis, Benedikt M. Scheuring, Wilfried V. Liebig, Andrew N. Hrymak, Frank Henning

**Affiliations:** 1Polymer Engineering, Fraunhofer Institute for Chemical Technology ICT, 76327 Pfinztal, Germany; 2Lightweight Design, Institute of Vehicle Systems Technology, Karlsruhe Institute of Technology, 76131 Karlsruhe, Germany; 3Hybrid and Lightweight Materials, Institute for Applied Materials, Karlsruhe Institute of Technology, 76131 Karlsruhe, Germany; 4Department of Chemical and Biochemical Engineering, Western University, London, ON N6A 3K7, Canada

**Keywords:** polycarbonate, glass fiber, LFT-D, composite, fiber length, mechanical properties, specific mechanical energy, process parameter selection

## Abstract

Long-fiber thermoplastic (LFT) materials compounded via the direct LFT (LFT-D) process are very versatile composites in which polymers and continuous reinforcement fiber can be combined in almost any way. Polycarbonate (PC) as an amorphous thermoplastic matrix system reinforced with glass fibers (GFs) is a promising addition regarding the current development needs, for example battery enclosures for electromobility. Two approaches to the processing and compression molding of PC GF LFT-D materials with various parameter combinations of screw speed and fiber rovings are presented. The resulting fiber lengths averaged around 0.5 mm for all settings. The tensile, bending, Charpy, and impact properties were characterized and discussed in detail. Special attention to the characteristic charge and flow area formed by compression molding of LFT-D materials, as well as sample orientation was given. The tensile modulus was 10 GPa, while the strength surpassed 125 MPa. The flexural modulus can reach up to 11 GPa, and the flexural strength reached up to 216 MPa. PC GF LFT-D is a viable addition to the LFT-D process, exhibiting good mechanical properties and stable processability.

## 1. Introduction and State-of-the-Art

Long-fiber thermoplastic (LFT) materials encompass a wide variety of polymer matrix systems in combination with reinforcement fibers. LFTs are compounded on a variety of machines following different concepts. LFT materials can be bought in the form of stick granules that are pre-loaded with reinforcing fibers, as well as compounded on-site directly from polymer granulates and chopped glass fiber or glass fiber rovings. The main processing routes of compounded LFT semi-finished materials are injection or compression molding. Ning et al. [1] delivered a good overview of the state-of-the-art in LFT processing and researched material systems, as did Rohde-Tibitanzl [2] and Gandhi [3].

### 1.1. Motivation

Most manufacturers specify a typical application range for automotive parts between −20 °C and 80 °C. Typical polymers used in LFT-D processing have their glass transition temperature, $T_{g}$, in precisely this range (PA6 approximately 60 °C, PP approximately 10 °C). From this point of view, the use of amorphous PC with a $T_{g}$ of around 150 °C is very promising.

Polycarbonate is difficult to ignite and inherently flame retardant (FR) to a UL-94 V2 rating and needs only very little additives to meet higher standards, potentially offsetting the slightly higher material costs. Lin et al. found that adding single-digit amounts of FR additives to PC GF was sufficient to achieve a UL-94 V0 rating [4]. Levchick and Weil studied the FR requirements of PA6 and PC matrix systems. Depending on the fiber mass fraction ($w_{F}$) and FR additive used, up to $w_{A}$ = 35% additives need to be mixed into the PA6 matrix [5,6].

Looking at current developments in electromobility, with the industry looking to substitute metal battery housing structures with composite solutions, the property profile of PC LFT-D might be a suitable solution.

### 1.2. State-of-the-Art

For more than 20 years, direct LFT (LFT-D) processing technology has been available from Dieffenbacher, and this article focused on the mechanical characterization of sample plates manufactured in compression molding from LFT-D compounded on a Dieffenbacher line. A schematic depiction is given in Figure 1.

It comprises two Leistritz co-rotating twin screw extruders (TSEs). The first TSE compounds the matrix material and conveys a melt film to the second TSE via an open film die. Continuous fiber rovings are fanned out and added into the second TSE at the same time and placed as the melt, which is pushed through the fiber for initial impregnation. The fibers and melt are wound around the screws and continuous rovings sheared into fiber bundles. These fiber bundles are broken up, and single filaments are homogenized in the matrix material over the processing length of the second TSE. The result is a plastificate comprising molten polymer and fibers also called “log” [7], “strand” [8], “extrudate” [9], “initial charge” [3], or “charge” [10,11] in the known state-of-the-art works.

The following state-of-the-art focused on works conducted with this setup. All of the relevant literature showed the throughput of the polymer, screw speed, and amount of rovings to be the decisive parameters for the resulting fiber mass fraction and quality of the product. It is these three parameters that can be adjusted momentarily and govern the LFT-D process. It is not possible to set this parameter triad independently of each other while fixating on the total throughput and fiber mass fraction. Further important parameters are the screw design and temperature.

Throughout the literature, it is unclear which parameter has the greatest influence, presenting results seemingly dependent on trial design [2]. Works conducted with chopped fibers can only be adapted with great caution as one of the fundamental aspects of the process, the coupling of fiber throughput to screw speed, is not an issue.

The most-commonly investigated materials are polypropylene (PP) and polyamide 6 (PA6) reinforced with glass fibers (GFs) and carbon fibers (CFs) in varied fiber mass fractions. Tröster presented the development of PP GF for the LFT-D process. In his work, the second TSE was set to run at maximum filling degree [12]. Henning et al. presented the opportunities of the LFT-D process for automotive applications, highlighting potential advantages over glass mat thermoplastic materials (GMTs) [13]. Geiger et al. presented the use of acrylonitrile butadiene styrene (ABS) GF and styrene acrylonitrile resin (SAN) GF in the LFT-D process, as well as a fiber cutter to improve the dispersion of the fiber bundles for better surface qualities [14,15]. Radtke presented extensive characterizations of the fiber length distributions and corresponding orientations in LFT-D from a PP GF produced with an inline fiber cutter [16]. Smith et al. presented a study on the mechanical performance of CF and GF PA66 LFT-D materials and hybrid materials. They went with a constant screw speed and increased the roving amount to achieve higher fiber fractions [17]. Rohan et al. presented a mechanical study of PA6 CF produced at various screw speeds and found that the mechanical properties suffered at lower screw speeds. Material inhomogeneity in the form of fiber bundles was observed for all screw speeds [18]. Dahl et al. presented the effects of the TSE processing parameters on the performance of PA6 CF. They found insufficient fiber dispersion and wetting for screw designs without any mixing elements. Except for impact properties, the mechanical performance suffered accordingly. Observations were made regarding the TSE torque, which is directly related to the power consumption and, thus, the operating costs of the TSE. Screw designs with mixing elements caused higher torque, as did low screw speeds [10]. McLeod et al. reported on a benchmark of LFT-D compression vs. LFT-D injection molding for a PP GF and a study on the fiber lengths in the plastificate [7]. Hümbert wrote about the influence of screw design and speed on PA6 GF LFT-D. He noted the importance of good fiber dispersion over the fiber length for good mechanical properties. He also found that, while a high shear rate damaged the fiber length, as expected, this effect only occurred for high degrees of fill in the TSE [19].

There are two main approaches to setting the processing parameters in LFT-D. The first approach was described in Kohlgrüber for extrusion in general and adapted by Tröster, among others, for the direct extrusion of continuously fed fibers. The goal was to run the extruder at a high torque and low screw speed, without exceeding the machine limits. The idea was to minimize the shear stress on the fibers [12,20]. Shear stress, a key figure in extrusion, can be expressed by the specific mechanical energy (SME) shown in Equation (1) [20].
(1)$$SME=\frac{2\cdot \pi \cdot n\cdot M_{D}}{\dot{M}}$$
where *n* represents the screw speed in rpm, $M_{D}$ is the extruder torque in Nm, and $\dot{M}$ is the total LFT-D throughput in kg/h. A correlation between the SME and fiber length for LFT materials was shown, for example, by Inceoglu et al. for chopped fibers in PA6 twin screw extrusion [21]. Another way to set the parameters is fixing the fiber mass fraction $w_{F}$ and polymer throughput $m_{polymer}$ and calculating the screw speed and roving amount via Equations (2) and (3) beforehand.
(2)$$w_{F}=\frac{m_{fiber}}{m_{fiber}+m_{polymer}}$$
where the fiber throughput $m_{fiber}$ in kg/h is defined by Equation (3).
(3)$$m_{fiber}=n_{rov}\cdot Tt\cdot n\cdot v_{intake}$$
where $n_{rov}$ is the roving amount, *n* is the screw speed, and Tt is the linear density of the fibers in tex (g/km). $v_{intake}$ in m/rpm is an empirically evaluated factor that is dependent on the matrix material, the amount of rovings, as well as the screw speed. It describes the length of the fiber drawn into the extruder per rotation and is evaluated by marking a certain length of fiber and measuring the time it takes to draw it into the TSE [12]. In other works, this evaluation was performed via a scale [2] or an optical measuring device [22]. The fiber mass fraction in LFT-D compounds is often abbreviated by giving the reinforcement type (e.g., GF) and a number representing $w_{F}$ in percent (e.g., 40).

One aspect of process optimization is the characterization of the mechanical performance of plates molded from LFT-D. A stand-in for mechanical performance (e.g., tensile, bending, etc.) is fiber length, directly related to the mechanical performance of fiber-reinforced materials. Various model predictions regarding the increase of the mechanical properties with increasing fiber aspect ratio (and thus, fiber length) are available [23,24,25]. While the tensile modulus increases with increased fiber mass fraction, the tensile strength finds a maximum, where the increasing fiber mass fraction cannot be dispersed in the matrix properly, causing faults and decreasing the properties. Thomason reported a maximum of tensile strength at around $w_{F}$ = 40% for PP GF materials and predicted a maximum between $w_{F}$ = 40% and $w_{F}$ = 50% for PA6 GF materials [26,27]. Reported upper limits for PC LFT-D were not found in the literature. Investigations of the fiber length in LFT-D products are available. The reported number average fiber lengths span from 1 mm (PA6 GF30) [19] 20 mm (PP GF40) [7] to as high as 40 mm (PP GF30) [12]. To measure fiber length, there are multiple methods available. All rely on removing the matrix material. Then, a sample of the remaining fiber skeleton is taken for measurements. Here, a variety of methods can be deployed. Most rely on sub-sampling once the matrix is removed; the sub-sample is then counted, usually via an image analysis algorithm. Goris et al. presented a novel method in a comparative study with existing methods. They found that, while no uniform standards exist, the FASEP fiber length measurement is currently the only commercially available solution capable of delivering repeatable measurements by considering entire sample quantities without having to sub-sample and potentially skewing the measurements [28].

Measurement results can be displayed as a histogram of all fibers *N* measured divided into *n* categories. Each category has $N_{i}$ fibers with length $l_{i}$.

The usual key figures are a number average value $l_{n}$:(4)$$l_{n}=\frac{\sum_{i=1}^{n}\left(l_{i}\cdot N_{i}\right)}{\sum_{i=1}^{n}\left(N_{i}\right)}$$
and a weight average value $l_{w}$:(5)$$l_{w}=\frac{\sum_{i=1}^{n}\left(l_{i}^{2}\cdot N_{i}\right)}{\sum_{i=1}^{n}\left(l_{i}\cdot N_{i}\right)}$$
which considers longer fibers [2]. Reinforcing materials with discontinuous fibers require load transfer from the matrix into the fiber via interfacial shear forces. The goal for any composite material is to fracture the reinforcing fiber before pulling it out of the matrix. To achieve this, the critical fiber length $l_{crit}$, described by Kelly and Tyson, has to be exceeded [25].
(6)$$l_{crit}=\frac{\sigma_{f}\cdot r}{\tau_{y}}$$

The tensile strength of the fiber is $\sigma_{f}$; the fiber radius is *r*; the interfacial shear strength is $\tau_{y}$. The reported interface strengths between 22 and 34 MPa for PC GF composites can be found in Nam et al. [29], Uematsu et al. [30], and Wongpajan et al. [31].

While the fiber length is directly responsible for the mechanical performance and a primary optimization target, the dispersion of fiber bundles is of importance as well. Dry, undispersed fiber rovings are a common occurrence in LFT in general, as they are in LFT-D processed from continuous rovings. The phenomenon was reported by various sources. Uawongsuan reported poorly dispersed bundles negatively affecting the mechanical performance of samples produced in a direct process versus samples produced in a comparable process using pellets [32]. Tröster described the dry fiber bundles in the LFT-D plastificate and their subsequent propagation in parts after molding [12]. Hümbert noticed dry fiber bundles especially for low screw speeds in LFT-D [19]. Hirata used a TSE to produce LFT granulates. Fiber bundles in the produced granulates were counted and related to the processing parameters [33].

### 1.3. Outline

In this work, we present the processing and fiber reinforcement of an amorphous PC matrix that is novel to the LFT-D process. No definitive state-of-the-art approach to the parameter selection for processing LFT-D materials was found. Two different approaches to LFT-D parameter selection and part production are presented, applied, and evaluated. The fiber mass distribution, as well as fiber lengths in molded plates are discussed, and one parameter set was chosen for further characterization. The mechanical properties of PC GF LFT-D are presented here for the first time.

## 2. Materials and Methods

All trials were conducted with a Makrolon 2405 PC provided by Covestro and StarRov 853 continuous glass fiber direct rovings provided by Johns Manville. LFT-D processing was conducted at Fraunhofer ICT on a LFT-D machine manufactured by Dieffenbacher GmbH Maschinen- und Anlagenbau, Eppingen, Germany. It comprised two Leistritz TSEs by Leistritz AG, Nürnberg, Germany. The first TSE was a Leistritz ZSE 40HP GL/32D with 55 kW nominal power. The second TSE was a Leistritz ZSE 40 GL/14.5D with a nominal power of 27 kW. The rectangular plastificate die on the second TSE was 75 mm wide and set to a height of 29 mm. The screw setups are shown in Figure 2 and Figure 3.

The polymer was dosed via gravimetric dosing scales from Brabender GmbH & Co. KG, Duisburg, Germany. All heating zones in both extruders were set to 300 °C, complying with the manufacturer’s processing advice. A DYL 630/500 parallel-guided hydraulic press with an effective usable press force of 5000 kN manufactured by Dieffenbacher GmbH Maschinen- und Anlagenbau, Eppingen, Germany, was used. A 400 mm × 400 mm polished steel mold with a diving edge was used to produce the plates.

### 2.1. Choosing Processing Parameters

Overall, the LFT-D material throughput per hour was fixed to match the part weight and press cycle time to avoid waste. Different desired fiber mass fractions, GF30, GF40, and GF50, were chosen. A subsequent focus was set on investigating GF40 materials, as a maximum of the mechanical properties can be expected here and as it is a very common value for industry application. The amount of fiber rovings is often determined by local conditions (e.g., fiber storage space or quantities available). An upper limit for how many rovings can be deployed in parallel is given by the size of the fiber intake of the extruder, as well as the width of the waterfall polymer nozzle of the first TSE. With the overall throughput and desired fiber mass fraction set, the fiber throughput needed to be addressed. Fiber throughput is always a product of screw speed and roving amount and calculated via Equation (3). For a fixed polymer throughput of 30 kg/h, the resulting fiber mass fraction is illustrated in Figure 4. This is a visualization of Equations (2) and (3). Perfectly matching the fiber mass fraction is not possible as the screw speed and roving amount can only be adjusted in integers. These considerations are valid for both approaches to LFT-D processing.

For the first parameter sets, process parameters were set on the torque or volume limit in accordance with Tröster [12]. To find this limit, the screw speed was decreased step-by-step, and the roving amount used was set as discussed. The material intake zone in the second TSE was monitored for overflow, as well as the torque limit. The resulting parameter combinations and total throughput are shown in Table 1. The torque limit was not reached for any combinations shown.

Additional parameter sets were calculated using fixed steps of the roving amount. Matching the constructive framework of the LFT-D line, 24 fiber rovings were chosen as a center point. The roving count was varied from 4 to 28 and 20, respectively. The calculated parameter combinations are given in Table 2.

The press closing profile is given in Table 3 and was kept constant throughout all parameter sets. The molding time was set to 30 s. The press force was set to 3200 kN, equaling 20 MPa of pressure on the filled plate, a value commonly reported for compression molding of LFT-D materials [34,35]. The mold temperature control was set to 100 °C and 105 °C for the upper and lower half, respectively. The mold will heat up during the beginning of molding via hot plastificates and remain stable at 115 °C and 120 °C throughout the processing. TSE torque was monitored and logged during the trial runs.

The manufacturing took place in two stages. The parameter sets V1 through V16 were produced initially. From this initial trial run, a parameter set was chosen for detailed mechanical characterization. The deciding criteria for this pre-selection was the fiber length, as well as a 40% fiber mass fraction. In a second trial run, the parameter sets V30 through V33 were produced. All parameter sets were sampled to determine the fiber mass fraction and fiber length.

### 2.2. Specimen Preparation

All samples were taken from 400 mm × 400 mm plates with a thickness of 4 mm. For all specimens, a distinction was made between the charge and flow area. Distances of >50 mm to the plates edges were respected to avoid the influences of flow phenomena on the fiber orientation. Samples were taken in different orientations with respect to the flow path, noted in the sample designation in Figure 5. For the characterization of the fiber mass fraction and fiber length, both the charge and flow area were sampled. For all other characterizations in the following experimental setups, the samples were cut from the pressed plates using a water jet system iCUTwater smart from the company imes-icore GmbH, Eiterfeld, Germany, with a pressure of 1500 bar, a cutting speed of 900 mm/min, and a flow rate of 250 g/min.

#### 2.2.1. Thermographic Analysis

Thermographic analysis (TGA) was performed on a TGA801 by LECO Instrumente GmbH, Mönchengladbach, Germany. Samples were burned off at a temperature of 650 °C for 10 h. The initial specimen mass, mass loss, and the mass of the residual fiber material were measured. $w_{F}$ was calculated from these measurements.

#### 2.2.2. Fiber Length Measurement

The fiber length measurements were carried out via FASEP by IDM Systems Dipl.-Ing. (FH) Helga Mayr, Darmstadt, Germany [36]. Entire samples from the TGA were used to conduct fiber length measurements. These samples were dispersed in water. Stirring the suspension in an ultrasonic bath was performed to facilitate dispersion. The entire sample was measured via FASEP over a couple of scans.

#### 2.2.3. Micrographs

Sample preparation was carried out in 7 polishing steps, described by Sharman et al. [37]. Since PC and glass have the same refractive index, no fiber detection was possible on the polished surfaces. Therefore, in a further step, the surface was etched with acetone for 10 min and then dipped in ethanol. This made the surface of the PC rough, while the glass fibers remained unchanged, allowing characterization. In addition, the contrast and brightness of the recorded images were edited in GIMP to increase the visibility of the fibers.

### 2.3. Mechanical Testing

#### 2.3.1. Tensile Test

Tensile tests were performed in accordance with DIN EN ISO 527-4 on a ZwickRoell GmbH & Co KG, Ulm, Germany, universal testing machine with external extensometer multiextens and a load cell with 20 kN at a travel speed of 2 mm/min. In order to test more samples per plate and to clearly separate samples between charge and flow area, the geometry was adjusted to 200 × 15 mm. To verify the directional material behavior, specimens with 0°, ±45°, and 90° orientations were tested. In order to examine the transition from 0° to −45° more precisely, additional specimens with −11.25° and −22.5° were tested. At least eight specimens were tested for each orientation, and only specimens failing within the valid gauge length were used for further evaluation. At least five valid attempts per orientation could be achieved.

#### 2.3.2. Bending Test

To test a larger volume, the bending tests were carried out as four-point bending based on DIN EN 14125 ISO on a ZwickRoell universal testing machine with a 2.5 kN load cell. Deflection was measured using an extensometer in the middle of the samples. The specimen design and test setup were selected according to Class-1, discontinuous fiber-reinforced plastics. Specimens with orientations of 0°, ±45°, and 90° were tested.

#### 2.3.3. Puncture Impact Test

The puncture impact test based on DIN EN ISO 6603-2 was carried out on an FW Magnus 1300 device from Coesfeld GmbH & Co KG, Dortmund, Germany to determine the impact sensitivity of the material system. Circular specimens with a diameter of 60 mm were used. The positioning of the specimens was performed with the help of a centering ring and collets. On the basis of preliminary tests, an impact energy of 25 J was selected for the charge and flow area. For each area, 17 samples were tested. The extra weight used was 4.9 kg, and the force signal was recorded using a 22 kN load cell. To investigate the fracture behavior in more detail, further puncture tests were carried out at different energies. Starting from the originally tested energy for the respective range, the energy was reduced by 2.5 J every two specimens. Thus, the fracture behavior could be observed over a larger energy span.

#### 2.3.4. Charpy Impact Test

The impact strength test was carried out in accordance with DIN EN ISO 179-1/1fU. The test specimens were 80 mm long and 10 mm wide. The support width was 62 mm. The test specimens were positioned narrow-sided in the machine. The HIT5.5P pendulum impact tester from ZwickRoell with a maximum working capacity of 5.5 J was used for the tests. The corrected work in joules was then documented for each specimen, and the Charpy impact strength was calculated according to DIN EN ISO 179-1/1fU. Specimens with orientations of 0°, ±45°, ±22.5°, ±11.25°, and 90° were tested.

#### 2.3.5. Dynamic Mechanical Analysis

Dynamic mechanical analysis (DMA) with PC GF40 and PA6 GF40 was performed on an Instron E3000 equipped with a temperature chamber and a 5 kN load cell manufactured by Instron GmbH, Darmstadt, Germany. For PC GF, as well as PA6 GF LFT-D, two specimens were tested in strain-controlled mode with a strain ratio of R = 0.3 and a frequency of 1 Hz. In order to analyze the temperature-dependent stiffness behavior, the testing was performed from a temperature of −30 °C until the specimens softened at a rate of 1 K/min. Since the highest influence of the fiber reinforcement was seen at a 0° orientation, only 0° specimens were tested here.

## 3. Results and Discussion

All trial points were evaluated for the fiber-related properties, fiber mass fraction, and length $l_{n}$. A parameter set for the additional production of PC GF40 plates was then selected on the basis of $l_{n}$ and secondary factors. Those plates were characterized in detail.

### 3.1. Fiber Mass Fraction Relating to Processing Parameters

The fiber mass fraction over all trial runs is shown in Figure 6. For the comparability over the entire range of GF30, GF40, and GF50, the measured fiber mass fraction was divided by the targeted fiber mass fraction. Data points in the box plot are grouped across the board by the targeted fiber mass fraction. In these groups, screw speed is shown ascending from left to right along with the corresponding roving amount and trial number.

Overall, Equation (3) underestimated the actual fiber throughput and $w_{F}$ accordingly. The actual $w_{F}$ were higher than calculated at around 42%. In the future, $v_{intake}$ should be determined per parameter set to match the desired fiber mass fraction precisely.

### 3.2. Fiber Length Relating to Processing Parameters and Processing Stability

The fiber length was chosen to be the key selection criterion for the first set of parameters, V1–V16. The results of the measurements are shown in Figure 7 for all parameter sets with GF40. The fiber lengths are short for an LFT-D material. In the literature, the differentiation between short and long fiber materials is drawn at an aspect ratio of one-hundred based on the works of Halpin and Pagano [38]. The aspect ratio is the quotient of the fiber length and diameter. The glass fiber used here had a diameter of 0.016 mm, putting the boundary between short and long fibers at 1.6 mm. While no significant differences can be seen for GF40, a slight increase towards higher screw speeds can be seen. The parameter set V6, with the highest mean $l_{n}$ of the first set, was chosen for detailed characterization in a repetition run named V33.

The fiber lengths at 16 evenly distributed positions were characterized. The fiber mass fraction and length $l_{n}$ in 16 sectors and a histogram of all fibers measured $l_{n}$ are depicted in Figure 8. Analogous to Figure 5, the charge and flow area are left and right, respectively. Both areas showed no differences in $w_{F}$ and $l_{n}$ and very little deviation in general, indicating that fiber damage during molding does not seem to play a role in PC LFT-D materials, as it does for other matrix systems.

Fiber damage during molding occurs when fibers freeze to the mold surface and are sheared by melt flowing in the middle of the plastificate during mold filling via fountain flow and is observed for compression and also injection molding [2,3]. Calculating from Equation (6) with literature values for $\tau_{crit}$ placed $l_{crit}$ between 0.5 mm and 0.8 mm, indicated by two red lines in Figure 8 in the histogram. Concluding from the histograms’ secondary y-axis, depending on the actual $l_{crit}$ for this exact material combination, between 10% and 25% of fibers passed $l_{crit}$. However, the literature suggests an $l_{crit}$ as low as 0.31 mm for PC GF compounds [31]. To check whether the load was transferred from the matrix to the fiber, micrographs of the fracture surfaces were made and discussed in Section 3.4.

### 3.3. Correlating Specific Mechanical Energy and Fiber Length

The TSE torque was monitored during all trials, and the SME was calculated according to Equation (1). The resulting SME and $l_{n}$ are displayed as a smoothed contour plot in Figure 9, SME left and $l_{n}$ right, over the parameter space defined by the roving amount and screw speed from Figure 4. The parameter combinations of roving amount and screw speed from Table 1 and Table 2 are indicated as black dots.

The SME was dependent on the screw speed and fiber mass fraction. The fiber mass fraction increased the TSE torque, as did low screw speeds. With a fixed polymer throughput, the overall throughput increased with the fiber mass fraction. Calculation wise, a higher overall throughput decreased the SME; however, the fiber load causing higher torques outweighed this. High screw speeds decreased the torque, but increased the SME per Equation (1). The increasing screw speed outweighed the reduction in torque regarding the calculated SME. On the relevant curve of $w_{F}$ = 40%, the SME increased from 15 kWh/kg (V16) to 22 kWh/kg (V6). Regarding energy consumption, operating at higher screw speeds was favorable as motor amperage was lower. This contributed to the decision for V6 (74 rpm and 20 rovings) as the parameter set for further investigation.

Per the state-of-the-art, the operating point would be on the very right of the parameter space, at low screw speeds [12,20]. While the differences in fiber lengths were not significant, the tendency shown here, longer fibers at higher screw speeds, does not agree with the state-of-the-art.

A correlation between the SME and fiber length as proposed by Inceoglu et al. can also not be confirmed at this point [21]. A possible explanation for this discrepancy is the existence of a minimal fiber length, where shear forces are not sufficient to break the fiber anymore [39,40].

Besides fiber shortening due to shear forces, another effect might be taken into account to explain this. Continuous rovings were drawn around the screw and were initially sheared between the screw and tapered housing diameter [12,41]. For high-viscosity matrix materials, a phenomenon has been reported, where stable “super-lattices” of fiber bundles exist, which travel through the extruder without being broken up. This would allow longer fibers in the form of bundles to be transported over certain distances before the bundles are dispersed [2,42,43]. In the amorphous matrix system of PC, fiber bundles were observed for all parameter sets. While hard to quantify, a general increase of the amount of fiber bundles was observed for lower screw speeds. Slower screw speeds demand higher roving amounts, increasing the surface area of the glass to be covered with the polymer melt in the intake zone of the second TSE at first contact. Higher shear forces, occurring at higher screw speeds, facilitate dispersive mixing, breaking up fiber bundles [20]. When the fiber bundles are broken up at the right time and place in the extruder, for example due to a beneficial combination of the shear force and residence time, fibers in the final product might be longer even when the shear forces are higher, and in theory, the fibers should be shorter. The overall uncertainty regarding the decisive process factors in the state-of-the-art might be attributed to this very complex environment.

### 3.4. Evaluating Micrographs

Figure 10 shows the micrographs of PC GF LFT-D from the charge area (a) and the flow area (b), with the flow direction pointing out of the plane in both cases. Figure 10a clearly shows the formation of a shell region at both edges of the plate, in which the fibers are oriented 90° to the flow direction. This can be attributed to the fact that the extrusion direction of the plastificate was also 90° to the later flow direction (cf. Figure 5). At the outer edges of the die, the fibers in the plastificate were aligned in this orientation as well. As soon as the plastificate came during the molding process into contact with the mold, these layers froze immediately, and the orientation was retained. Between the two shell areas, a core area was formed in which the fibers were aligned in the direction of the flow. This was due to the fact that the still-hot and -flowable interior of the plastificate was pressed into the cavity during the molding process, creating a flow front. The same orientation was obtained over the entire cross-section in the flow area (see Figure 10b). Since only the material from the interior of the plastificate, which underwent a flow movement during the process, reached the flow area, all fibers were oriented in the direction of the flow.

The evaluation of the fracture surface as, for example, shown in Figure 11, can be utilized to judge the viability of composite materials, especially whether the critical fiber length was reached. Analyzing the fracture surfaces, the following items can be indicative:Deeply staggered fracture surfaces hint at fiber fracture rather than matrix fracture. This can be observed as the entire depiction had a noticeable depth indicated by the darker and lighter areasMatrix material residue on fiber indicates a previous bond between fiber and matrix. Good examples of this can be observed in the right of the figure. These fibers look like they have been pulled out.In the load direction (here, the direction of view), short fiber stumps, as well as shallow holes can be observed in the very center of the figure, hinting at fiber fracture, rather than pull out.Clean fiber front surfaces also indicate fiber fracture, as when the fiber was pulled out, matrix material could be seen there.

While the measured fiber lengths $l_{n}$ were in the borderline range of the calculated critical fiber length $l_{crit}$, the analysis of the fracture surface indicated that $l_{n}$ surpassed $l_{crit}$ and reinforcing effects were at work even if the desired aspect ratio of >100 was undershot.

### 3.5. Investigation of Quasi-Static Material Behavior

In Figure 12, polar plots of the mean values for the tensile modulus (a) and strength (b) in different orientations for both the charge and flow area are shown. It is clearly visible that both properties hardly differ between the charge and flow area. The highest characteristic values were oriented at 0° (charge 10.6 GPa and flow 10.7 GPa) and the lowest at 90° for the flow area (5.6 GPa) and at −45° for the charge area (6.1 GPa). The 90° specimens in the charge area showed slightly higher stiffness (6.7 GPa) than in the flow area, which was confirmed in tensile testing (cf. Figure 12b). Both plots, tensile strength and stiffness, of both areas are very similar. Based on both characteristics in both areas, a main reinforcement effect of the fibers in a 0° orientation can be observed. The tensile strength, as well as the modulus of elasticity showed their highest values in this direction, indicating fibers aligning in the flow direction.

**Figure 11 polymers-15-02041-f011:**
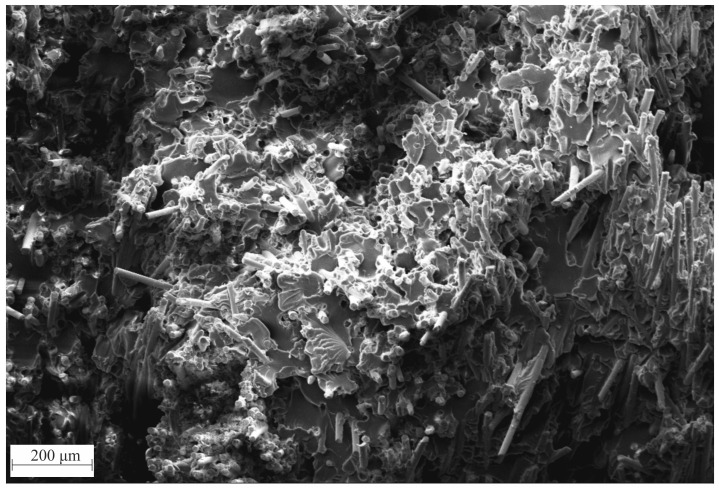
SEM micrograph of a fracture surface of a 0° PC GF40 specimen out of the flow area after the tensile test. Fractured glass fibers are embedded in polymer matrix.

Figure 13 shows the results of the four-point bending tests, again in two polar plots similar to Figure 12. The flexural modulus of elasticity in Figure 13a exhibits an anisotropic material behavior with its maximum in a 0° orientation. The difference between the charge and flow area was much stronger here than for the tensile properties (cf. Figure 12). While the flexural modulus in the charge area was 8.4 GPa at 0° and 8.1 GPa at 90°, it dropped by more than 50% from 11.1 GPa at a 0° orientation to 5.1 GPa at 90°. A similar situation can be observed in Figure 13b, where the charge area shows lower values at 0° (185.6 MPa) and a higher values at 90° (159.1 MPa) in comparison with the flow area (at 0° 216.1 MPa and at 90° 107.6 MPa). However, specimens from the charge area showed a significantly higher strength at 0° than at 90°, which means that the difference from the flow area was less pronounced than with respect to the stiffness. In a ±45° orientation, however, both areas showed very similar values in stiffness and strength.

### 3.6. Orientation and Area Dependency of Quasi-Static Mechanical Properties

The tensile and flexural properties exhibit strong, observable, orientation-dependent behavior in both the charge and flow area. In the flow area, this was due to the fiber orientation caused by eponymous material flow during compression molding. Testing showed that the orientation also occurred in the charge area, where the plastificate was placed after compounding. Since the original fiber orientation of the outer layer of the plastificate in the charge area was frozen upon contact with the mold, a shell structure where the fibers were oriented 90° to the flow could be detected. Confirming the micrographs shown in Figure 10 and discussed in Section 3.4 backed up the literature about the products of the LFT-D process. The initial fiber orientation in the plastificate was found responsible for the mechanical properties in the charge area. This fiber orientation in the shell and the core of the plastificate has been described by various authors, reporting fibers being oriented in the extrusion direction in the shell and at an angle of 60° in the core [3,8,12].

It is also striking that the properties of both areas differed significantly in the bending test, while they remained nearly equal in tensile testing. This can be attributed to the fact that the outer layers of the structure had a different influence on the tensile than the flexural properties. The load in the tensile testing was evenly distributed over the entire specimen volume. Since the amount of fibers in the shell was minimal compared to the amount of fibers within the core, the influence was minimal as well. Especially when the force was applied in a 0° orientation, the results for the charge and flow area were almost identical. In the 90° direction, the shell fibers showed a stronger influence on the tensile modulus. Since the fibers in the core structure practically no longer carried any load due to their “misalignment”, the influence of the shell fibers on the tensile modulus became greater (cf. Figure 12). This can also be seen in the stress–strain curves of the 0° and 90° orientations shown in Figure 14 with the corresponding scatter band.

The scattering range of the 90° specimens was significantly larger in the charge area than the flow area. All 90° specimens in the charge area had a higher stiffness and most of them a higher strength than in the flow area, additionally showing a lower elongation at break, which is a further indication that the fibers in the shell zone were loaded. Confirming this was the fact that, in the charge area, 0° and 90° specimens showed almost the same fracture elongation.

The minimal influence of the the outer shell fiber layers on the tensile modulus at 0° is illustrated in comparison to the bending modulus. The resulting flexural and tensile modulus for the flow area coincided well with deviations of a maximum of 6%, while in the charge area, the fiber orientation in the shell led to a sharp drop of 20.22% compared to the tensile modulus. This was due to the high loading of the shell fibers in the bending test. While the volume was uniformly loaded in the tensile testing, in the bending test, stress maxima occurred at the component boundaries. A schematic illustration of the discussed subject matter is given in Figure 15. Here, both load cases, tensile and bending, are shown with their respective stress distribution curve. The different microstructures, with the shell and core layer, in the charge and flow area are indicated.

Since the shell fibers in the charge area were positioned perpendicular to the fibers within the volume, these can act as a possible weak point in the case of bending. The loading in a bending test at a 0° orientation was thus mainly a loading of the shell fibers in the 90° direction, which greatly reduced the strength increase of the reinforcing fibers and could lead to cracking with subsequent failure. The flexural modulus of the charge region for 0° and 90° differed by 273.9 MPa, while the flexural modulus from 0° to 90° in the flow area dropped by 6.01 GPa (cf. Figure 13). This was again due to the stress distribution and loading of the shell fiber. While the edge fibers of the specimens with a 0° orientation can serve as a weak point, they carried most of the load when specimens with a 90° orientation were loaded. In this case, the edge fibers were oriented in the direction of the load and, thus, could use their maximum potential as long as complete wetting, as well as exceeding the critical fiber aspect ratio are assumed. The load in the bending case was not carried exclusively by the shell fibers, but also by the fibers inside the volume. The role of the fibers was, thus, reversed in the case of 0° and 90° orientations. If the edge fibers at 0° serve as possible weak points and the fibers in the volume as reinforcement, the edge fibers at 90° act as reinforcing components and the fibers inside the volume as possible weak points. In the case of flexural strength, there is no guarantee that failure will occur at the edge fibers. Even if the highest stresses occur there, local stress increases and thus failure can occur inside the volume and, thus, outside the influence of the outer fibers. The flexural strength was, therefore, not determined exclusively by the edge fibers. In the context of tensile strength, the edge fibers should again have no significant influence on the tensile strength because of the uniform stress across the entire cross-section. However, they could still serve as a weak point, depending on the load orientation.

Since the original fiber orientation of the outer layer of the plastificate was frozen in the charge area upon contact with the mold and this was 90° to the flow direction, it is not surprising that the bending properties at 90° to the flow direction were significantly better than those in the flow area.

### 3.7. Puncture Impact Test Results

In Figure 16, a representative curve of the puncture impact test in the charge and flow area is given. All curves can be found in Figure A1 in Appendix A. The area formed under the respective curves indicates the energy required to puncture the sample. In the upper right corner of the figure, the average and the standard deviation from all 17 tested specimen in both areas are shown. In addition, further curves for individual areas are shown in the Appendix A in Figure A1.

Both curves have a jagged course shape, typical for fiber-reinforced polymers. Once the fibers broke, a rapid drop in force was detected, increasing again, as soon as the next fibers were loaded. As soon as these broke again, the next drop occurred, and so on. Both curves in Figure 16 and all curves in Figure A1 show after a first steep increase of force (deformation 0 mm–1.25 mm), a first plateau (1.25 mm–3.75 mm), followed by a second, flatter, increase (3.75 mm–6 mm). A second plateau (from 6 mm–8 mm) and a sharp drop in the curve (approximately 8 mm–9 mm) follows. After this, the curve continues to flatten out until it breaks off at 0.25 kN. Even if the course of the curves is very similar, the charge area showed a higher breaking strength of 2.9 J on average. In order to better understand the fracture behavior, additional tests were performed at different impact energy levels.

Figure 17 shows a set of curves of a specimen from the charge area at the above-mentioned energy levels. In order to transfer the curve progression to the fracture behavior, additional images of the tested specimens are given at the bottom of the figure to track the propagation of failure. The first crack formed in the fiber direction, indicated by the red arrow “orientation of flow”. The second crack formed perpendicular to the fiber orientation for all specimens. For 5 J, 7.5 J, and 10 J impact energies, cracking occurred with minimal bulging at 7.5 J and 10 J. At 12.5 J and 15 J, the first fracture edges formed. At 17.5 J and above, the first breakout could be seen in one of two specimens. At 20 J, severe cracking occurred with minimal breakout in one of two specimens. From 22.5 J on, the specimen was punctured in three of four specimens.

A characteristic master curve shape could be identified, all samples showing the first plateau and second plateau where applicable. It can be assumed that the first plateau represents the energy required for hairline fracture formation in and perpendicular to the fiber direction. The course of the curve up to the maximum at the second plateau represents further fracture surfaces forming for higher impact forces where entire sections of the material are broken off. In the case of the first breakout for one of the specimens at 17.5 J, a sharp drop in the curve can be seen for the first time before it also progresses toward zero. The backward slopes of the curve towards the zero point are the load cell bouncing off of the specimen. They were not removed to clarify the end point of the load, but no longer represent a course of the load. The three specimens, which experienced a breakthrough from 22.5 J, show a steady curve without any return without regressing towards the zero point.

### 3.8. Charpy Impact Results

The evaluation of Charpy impact testing (Figure 18) also showed an anisotropic curve (cf. Figure 12 and Figure 13). This time, there was hardly any difference in the orientations ±22.5°, ±11.25°, and 0° for both the charge (44.4 kJ/m^2^–45.5 kJ/m^2^) and the flow area (43.1 kJ/m^2^–44.4 kJ/m^2^). At the transition to a ±45° orientation, there was already a clear drop for both areas, which tended to be stronger in the flow area. The most-significant difference in the ranges was seen in a 90° orientation; while the impact strength in the charge area was only slightly reduced (to 31.3 kJ/m^2^), a significant drop in the flow area (to 19 kJ/m^2^) can be seen.

The Charpy impact strength plot also explains the only slightly different results in puncture impact testing. The slightly lower impact strength in the flow area can be attributed to the proportion of 90° cracks. They showed significantly lower energy absorption with respect to the Charpy results. Since all samples had a crack parallel to a 0° and one to a 90° orientation, the 90° crack seemed to provide the same retention of the curve and the 0° crack for the slightly deeper run of the samples from the flow area.

### 3.9. Comparing Thermoplastic LFT-D Materials in Dynamic Mechanical Analysis

Figure 19 shows the DMA results of PA6 GF40 and PC GF40, both produced with the same screw configuration on the same machine. The curves of the storage modulus and the tan δ (at the peak of which $T_{g}$ is defined) over temperature are shown. Two significant differences can be clearly seen. The PA6 GF LFT-D showed an increase in tan δ at approximately 20 °C and reached a temporary maximum at approximately 68 °C, then dropped again. Simultaneously, the storage modulus showed a considerable drop in this area, which illustrates the effect of $T_{g}$. It can be seen that, even after $T_{g}$, there was still a certain stiffness in the material due to the crystalline phase. In comparison, PC GF LFT-D showed only a slight reduction in the storage modulus in the same temperature range. Once the temperature approached $T_{g}$ at 148 °C, complete softening of the material occurred, and no further testing was possible; the storage modulus dropped to almost 0 GPa.

The substantial variation of the storage modulus over temperature illustrates the potential benefits that structural components made from PC-based composites could have in comparison to the currently widely used semi-crystalline-based composites. For example, the stiffness of PC GF LFT-D due to the high $T_{g}$ over the relative application range in the automotive sector (−20–80 °C) was only affected by a drop of 5.4%. In contrast, the PA6-based composite suffered a severe drop in the storage modulus of 32.4% in the same temperature range as the $T_{g}$ of PA6 lies at 68.3 °C. As $T_{g}$ of PA6 is highly dependent on the moisture uptake of the polymer, this effect can be even worse. This comparison clearly showed the great potential of glass-fiber-reinforced polycarbonate (amorphous thermoplastic polymers in general) with regard to structural components.

## 4. Summary and Outlook

Processing PC in LFT-D is feasible and can be performed without changes to the overall process concept, and it is stable at various fiber loads and processing parameters in general. Processing at a low SME input does not have advantages, so the approach described by Tröster [12] and also Kohlgrüber [20] is not valid. Considering slightly higher fiber lengths combined with lower energy consumption, we suggest processing PC GF LFT-D at higher screw speeds. A suitable parameter set can be calculated by the means presented in this work. The fiber length distribution over the entire part was noticeably homogeneous in contrast to other LFT materials [3]. The mechanical properties were nonetheless comparable to other LFT-D materials and superior to injection-molded PC GF materials. Compared to other LFT-D materials, PC GF was characterized by superior temperature stability in a typical application range. The additional benefits of PC LFT-D such as the flame retardant properties or possible surface finishes should be investigated in the future.

## Figures and Tables

**Figure 1 polymers-15-02041-f001:**
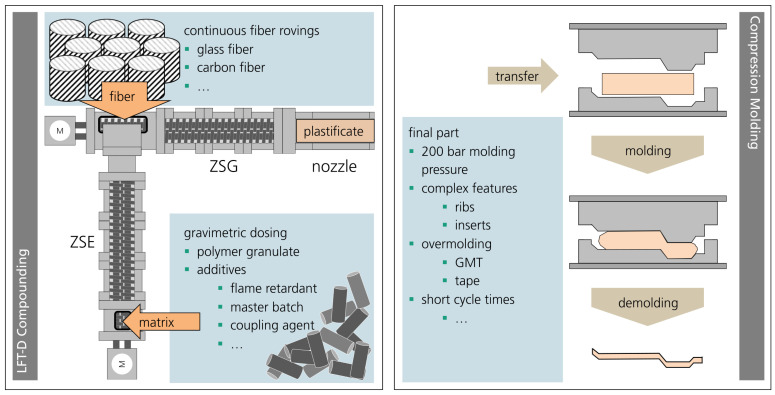
Process scheme for compounding (**left**) and compression molding (**right**) of LFT-D materials.

**Figure 2 polymers-15-02041-f002:**
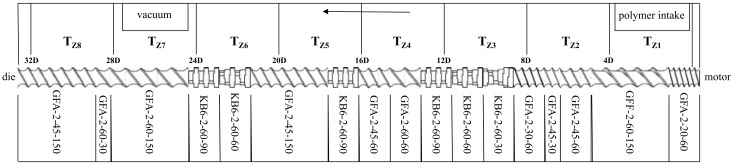
Screw configuration of first TSE. This screw facilitates the plastification of polymer granulates. Extrusion direction from right to left indicated by the arrow.

**Figure 3 polymers-15-02041-f003:**
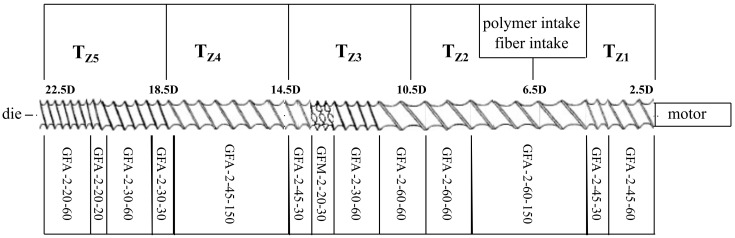
Screw configuration of second TSE. This screw design featured one mixing element (GFM) and was considered a low-shear setup. Fiber reinforcement was incorporated into the molten matrix material over the course of the screw. Extrusion direction from right to left indicated by the arrow.

**Figure 4 polymers-15-02041-f004:**
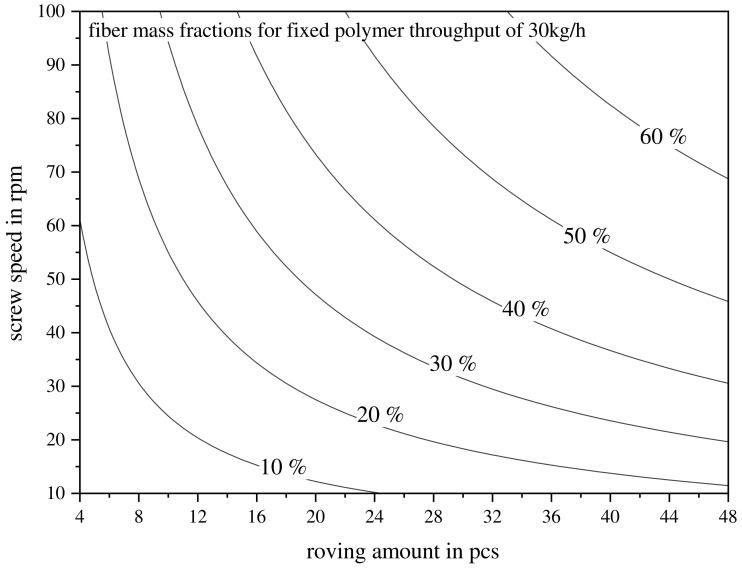
Relation of screw speed and continuous roving feed to resulting fiber mass fractions from 10% to 60% for a fixed polymer throughput of 30 kg/h in LFT-D.

**Figure 5 polymers-15-02041-f005:**
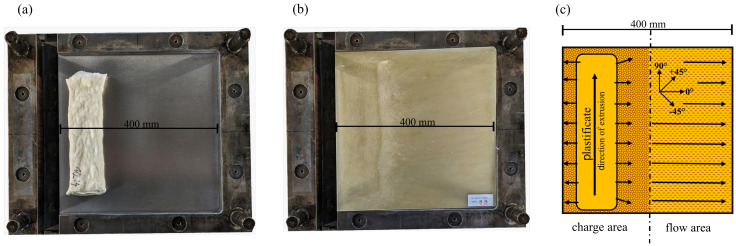
Representation of the molding process with (**a**) polished steel mold with plastificate in charge position and (**b**) polished steel mold with molded plate and (**c**) schematic of the square plate mold with charge area, flow path, and resulting flow area. The position of the LFT-D plastificate is shown, including the extrusion direction. Possible orientations of the sampling are indicated.

**Figure 6 polymers-15-02041-f006:**
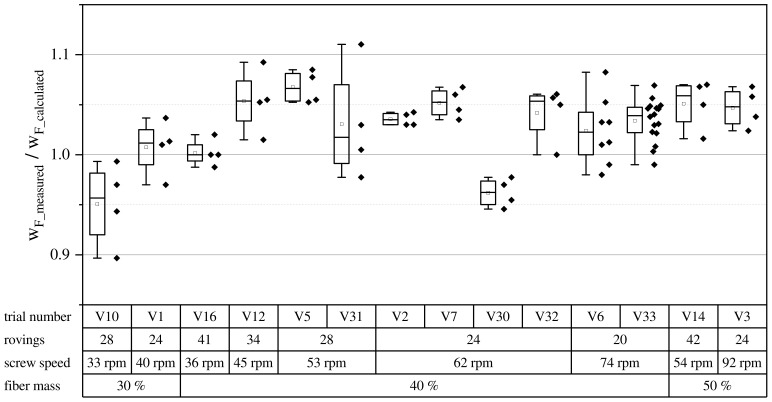
Normalized fiber mass over all fiber mass fractions and processing parameters. Each diamond represents one sample.

**Figure 7 polymers-15-02041-f007:**
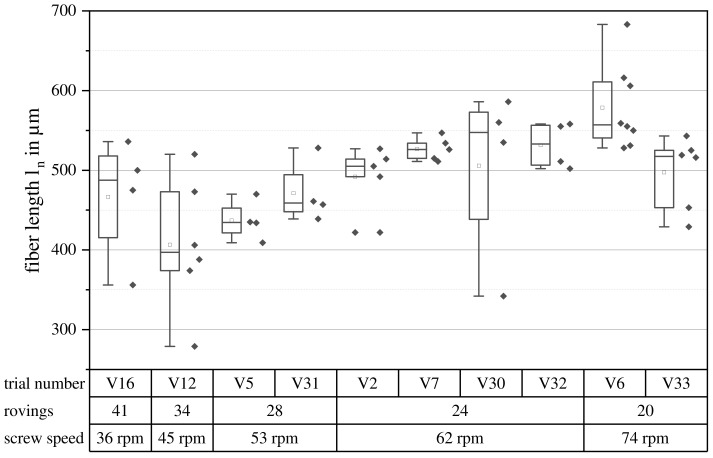
Fiber length $l_{n}$ over all parameter sets resulting in PC GF40. Screw speed is shown increasing from left to right. Each diamond represents one sample.

**Figure 8 polymers-15-02041-f008:**
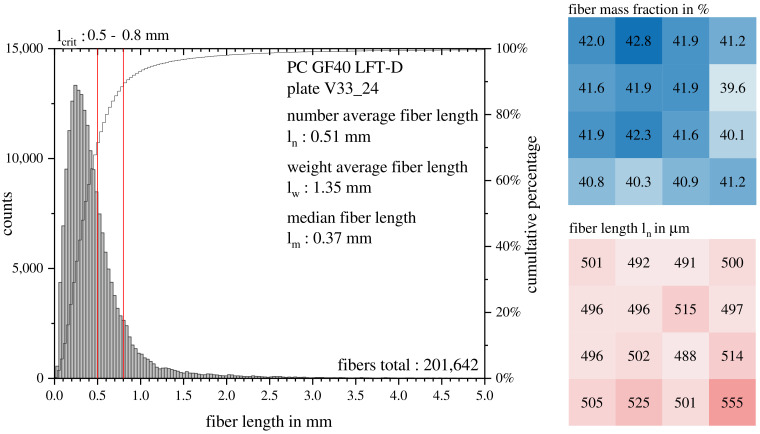
Fiber property evaluation of 16 samples distributed evenly over one plate from V33. A histogram of $l_{n}$ over all fibers measured is depicted on the left. Red lines indicating the possible $l_{crit}$. Fiber mass fraction is shown on the upper right and fiber length $l_{n}$ on the lower right for all 16 positions. Darker colors indicate higher fiber mass fraction or fiber length respectively.

**Figure 9 polymers-15-02041-f009:**
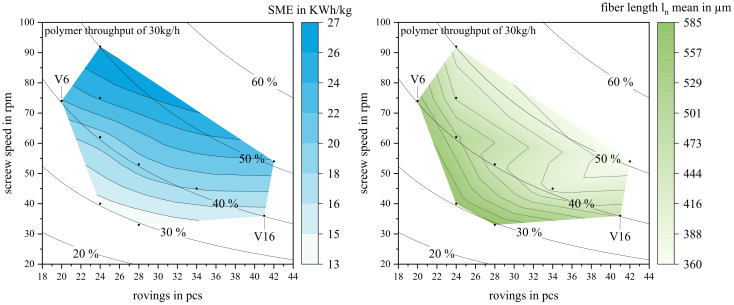
Parameter space of screw speed, roving amount, and resulting fiber mass fractions from 10% to 60%. Calculated SME left and measured fiber length $l_{n}$ right. Black dots indicate the various parameter sets. The parameter set V6 was used for all further characterizations.

**Figure 10 polymers-15-02041-f010:**
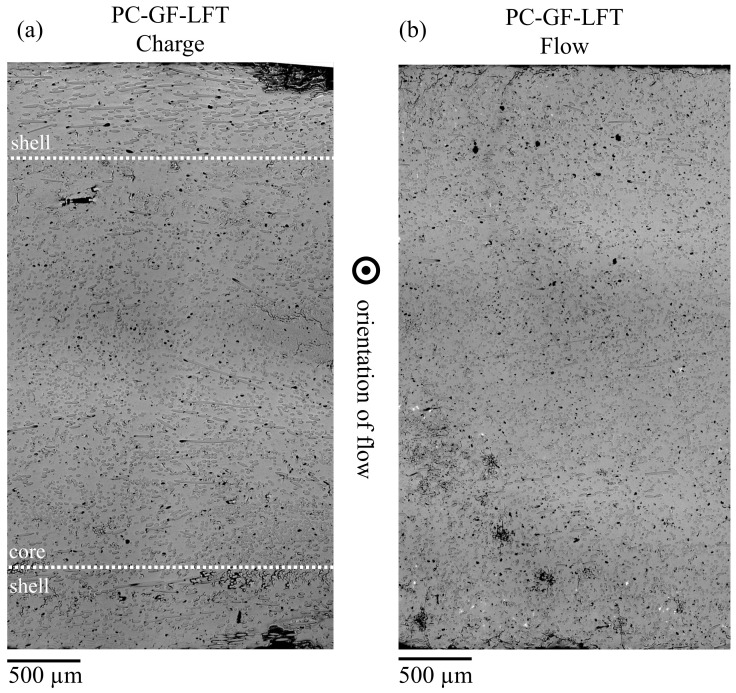
Micrographs of PC GF LFT-D from (**a**) the charge and (**b**) the flow area with the marked shell core effect in the charge area.

**Figure 12 polymers-15-02041-f012:**
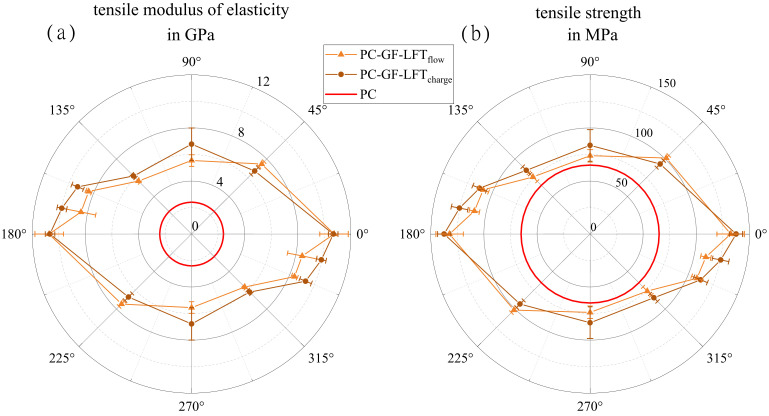
Polar plots of tensile modulus of elasticity (**a**) and tensile strength (**b**), each showing average properties in charge and flow area, respectively, with the associated scatter, in comparison with the results for pure PC taken from the data sheet.

**Figure 13 polymers-15-02041-f013:**
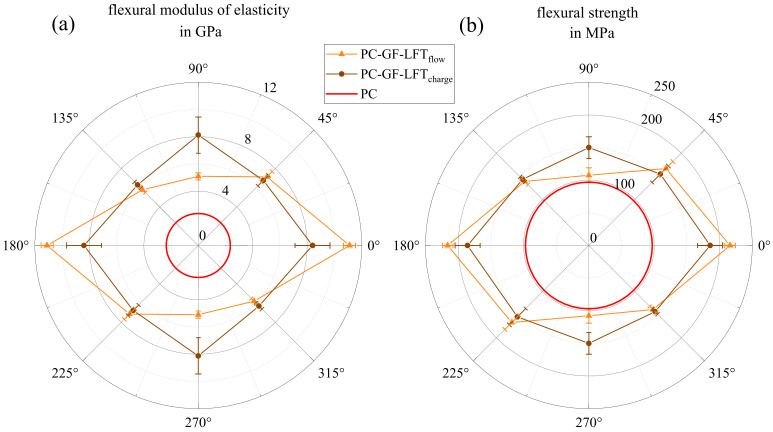
Polar plot of the results from the four-point bending test separated into charge and flow area in (**a**) the flexural modulus of elasticity and (**b**) the flexural strength, in comparison with the results for pure PC from the three-point bending test taken from the data sheet.

**Figure 14 polymers-15-02041-f014:**
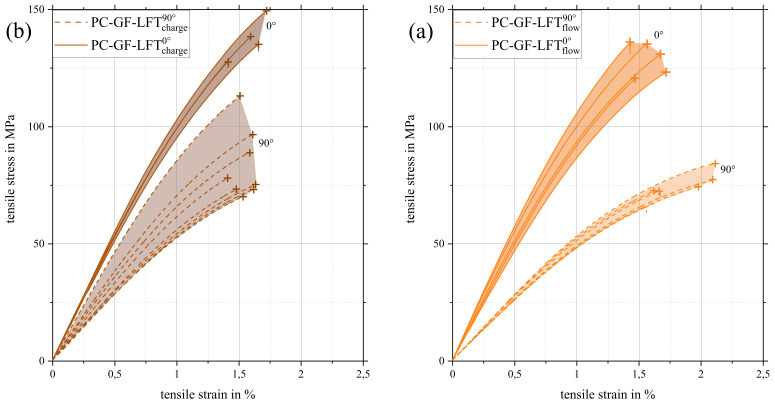
Stress–strain curves of tensile tests with corresponding scatter range of all valid specimens in 0° and 90° orientations separated into charge (**a**) and flow area (**b**).

**Figure 15 polymers-15-02041-f015:**
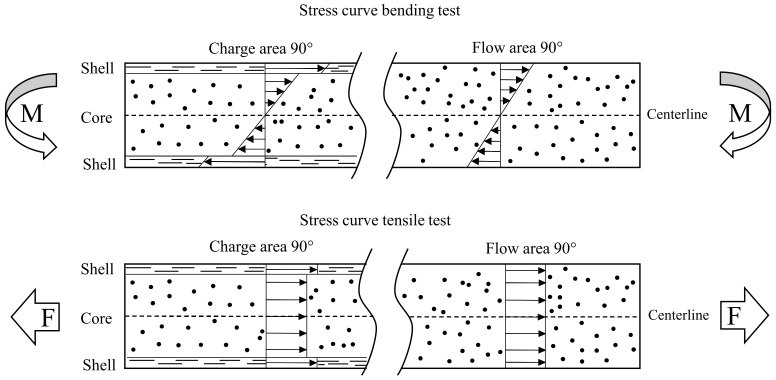
Schematic stress distribution and microstructure of specimens cut 90° to the flow direction. Schematically showing the shell and core zones in charge (**left**) and flow (**right**) area. Load cases for both bending (**top**) and tensile testing (**bottom**) are shown.

**Figure 16 polymers-15-02041-f016:**
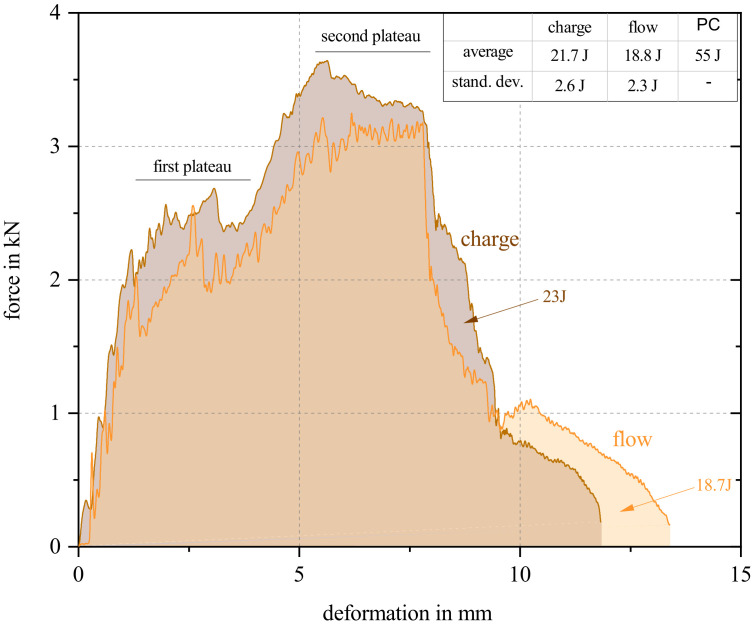
Representative puncture impact test curves of samples from charge and flow area. Average values, standard deviation of all tested samples, and the results for pure PC from the data sheet are shown in the table at the upper right edge of the figure.

**Figure 17 polymers-15-02041-f017:**
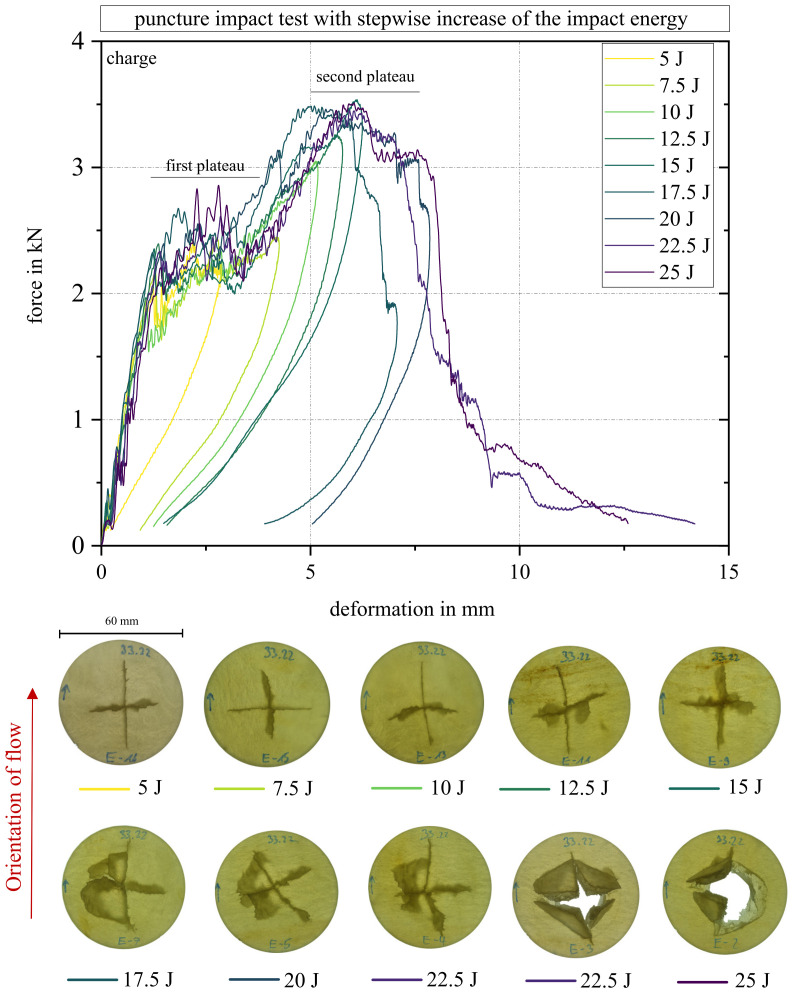
Curves of impact force over displacement and corresponding fracture patterns from 5 J to 25 J. Flow orientation equals fiber orientation and can be found in the fracture pattern.

**Figure 18 polymers-15-02041-f018:**
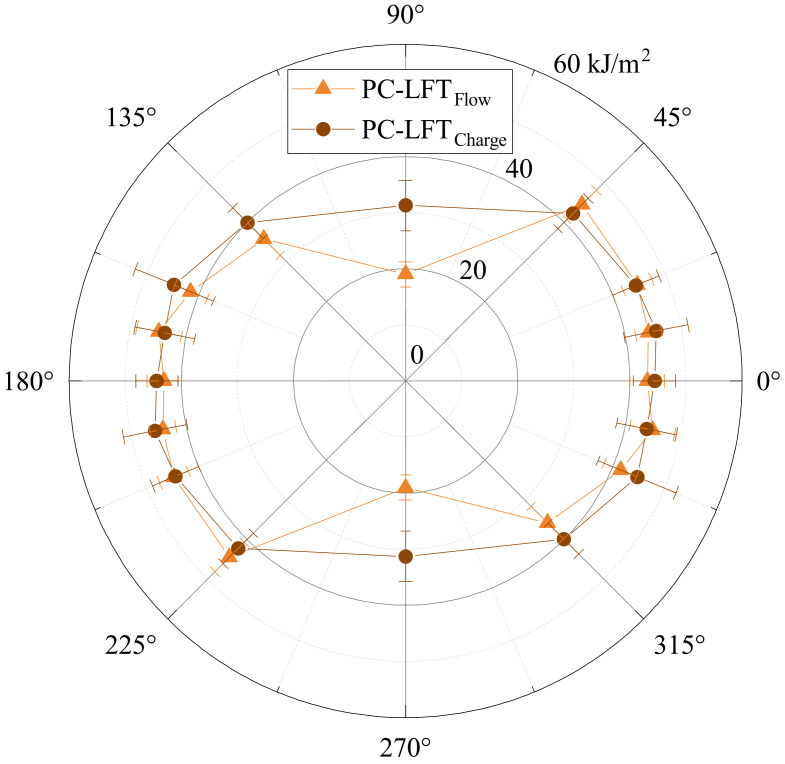
Charpy impact strength $\alpha_{cU}$ according to DIN EN ISO 179-1/1fU in a polar plot. Sampling in charge and flow area is shown separately.

**Figure 19 polymers-15-02041-f019:**
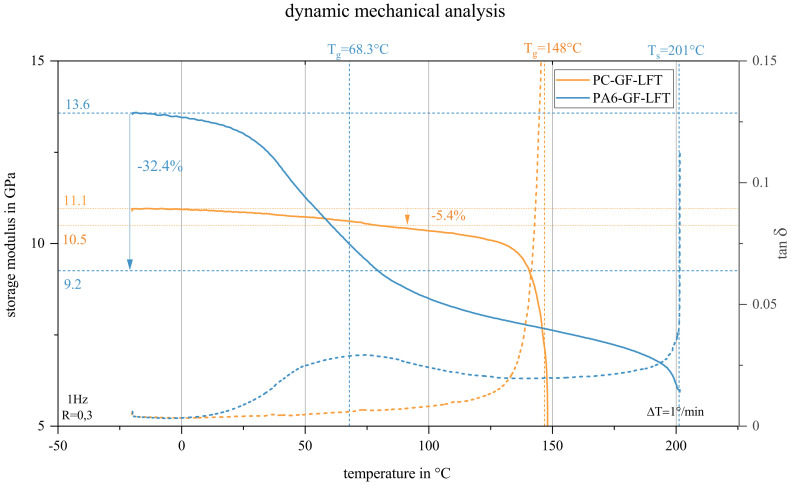
DMA curves of storage modulus and tan δ of a 0° specimen taken from the flow area of GF40 PA6 and PC plates.

**Table 1 polymers-15-02041-t001:** Operating points at maximum extruder fill level.

Trial #	Fiber Mass Fraction in %	Screw Speed in rpm	Rovings in pcs	Total Throughput in kg/h
V10	30	33	28	42.86
V16	40	36	41	50
V12	40	45	34	50
V14	50	54	42	60

**Table 2 polymers-15-02041-t002:** Calculated parameter sets.

Trial #	Fiber Mass Fraction in %	Screw Speed in rpm	Rovings in pcs	Total Throughput in kg/h
V1	30	40	24	42.86
V5	40	53	28	50
V31	40	53	28	50
V2	40	62	24	50
V7	40	62	24	50
V30	40	62	24	50
V32	40	62	24	50
V6	40	74	20	50
V33	40	74	20	50
V3	50	92	24	60

**Table 3 polymers-15-02041-t003:** Press closing profile.

Height Above Zero in mm	Closing Speed in mm/s
50	80
30	30
15	5
0	5

## Data Availability

All data are presented in the article.

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
