# Peer review of "Approaching Polycarbonate as an LFT-D Material: Processing and Mechanical Properties"

_polymers, 2023, doi:10.3390/polym15092041_

Round 1

Reviewer 1 Report

The manuscript under review studies the processing parameters  and the mechanical properties of PC reinforced with glass fibers munufactured by means of direct extrusion. The focus of his work is more technological than scientific, but it falls within the scope of "Polymers" and its contributions are relevant to the community.

From the methodological point of view, the work is sound and conclusions are backed by results. In this reviewer's opinion, it can be published provided authors address some points, which are listed below.

- One critical point is the fiber length. I particularly appreciate the histogram in Fig.8, which provides the full distribution. When a  distribution is skewed (and this is clearly the case), the median is a more reliable estimator than the mean value. It would be desirable to display both values in the lower right square of Fig.8.

- Given the fact that fiber length lies in the interval from 0.4 mm to 0.6 mm and the definition in Eq(6), it is important to consider whether the term LFT (long fiber thermoplastic) accurately describes the reinforcement. We all are aware of the usual sizes resulting from injection and extrusion, and therefore a short discussion would be welcome.

-From a formal point of view, acronyms such as FASEP (although usual), should be explained the first time they appear in the text.

-Charpy test is widely performed in the industry, but its scientifical value is limited. Results should be completed with a better measurement of toughness, at least in Mode I. This is just a recommendation for a future work.

-Lastly, it would be highly desirable to add the results obtained from tests performed on unreinforced polycarbonate in Figs. 12, 13, 14, 16, 17 and 18. This would provide a baseline set of properties, allowing a more complete understanding of the reinforcing effect of fibers and help discuss the effect of fiber orientation.

Even though the manuscript is generally well written, there are minor typos, such as "asses", "fractures surface" or "plastifkate". A final proofreading should be performed.

Author Response

Thank you for reading and commenting on the manuscript. We applied your comments and suggestions. I will go over it by paragraph according to your order below:

  • Median was calculated/identified and added to fig. 8
  • In this case the process and semi finished product is also called LFT-D. This denomination aides the categorization and shall be kept. Discussion and citation of works about aspect ratio and what constitutes the "long" in LFT is added to the manuscript. We are most certainly on a borderline case here. We feel that, especially discussing fracture surfaces, we can show that the reinforcement effect is there, irregardless of the aspect ratio.
  • FASEP does not seem to be an acronym. I included a reference to the paper where the developers introduced the methodology. Even there it is not explained.
  • noted, thank you!
  • We included baseline values for neat PC according to the datasheet provided by Covestro where applicable. Some testing proceedures are different and we explained where appropriate.

We did another round of proofreading and also improved on punctuation, spaces etc..

Reviewer 2 Report

The authors present an article with the title: Approaching Polycarbonate as a LFT-D Material: Processing and Mechanical Properties

The article is well-written, the language is clear and the methods are explained. There is only one thing which I did not understand and that is how the plates were prepared and therefore, what is the meaning of charge and flow area. I am actually not familiar with the LFT-D, which might be the reason.

In the WoS, I was not able to find an article on this type of PC composite preparation.

I believe that »a« in a title is wrong. (Actually, it is not me, it is Grammarly who shows it as wrong)

The abstract starts as an introduction and continues with methods, without showing any results. I suggest reducing the first part and adding results, which will increase the possibility of getting citations.

The introduction seems too long for me.

The word »coupler« in Figure 1 should most probably be a »coupling agent«.

m% is wrong according to ISO 80000: (Additional information, such as % (m/m) or % (V/V) shall not be beside the unit symbol %. The preferred way of expressing, e.g., the mass fraction of B, is wB = 78%. (the same goes for wt.% which is mostly used)

Figure 9: fibre length is in micrometers not meters.

The expression “loss angle” I have never heard at the DMA. I found it at the dielectric loss, which is actually very similar or the same. However, if one is talking in loss angle, it is delta, not tan delta. The authors are showing tan delta.

Author Response

Thank you for reading and commenting on the manuscript. We applied your comments and suggestions. I will go over it by paragraph according to your order below:

  • We have improved upon fig. 5 to supplement the explanaition of charge and flow area with actual pictures next to the scheme. The plastificate is compression molded wich forms two microstructurally distinct zones called charge(where the mass is initially positioned in the mold) and flow (where the material is flowing).
  • It seems that "an" is correct because of the phonetic sound of eL in abbreviations. Fixed title and some text passages accordingly.

  • Shortened a line in methods. Added fiber length results. Added key mechanical properties, tensile and bending. We feel, that the other mechanical properties need more in-depth discussion not appropriate for an abstract.

  • Shortened a couple of lines in "Motivation" and compacted the text. The introduction is the most comprehensive overview of works conducted with the LFT-D process known to me. While still long, I hope it will be a valuable ressource for readers.
  • Changed the wording.
  • The normative standard is very helpful- thank you for pointing that out. All relevant figures and tables were altered accordingly. Text was extensively corrected.

  • Re-converted the image. It now shows micro (before it was white on white)

  • Figure 19 was revised. Text was corrected accordingly

We revised further spelling mistakes and punctuation as well as spaces.